# Integrating unsupervised and reinforcement learning in human categorical perception: A computational model

Giovanni Granato[1,2]*, Emilio Cartoni[1], Federico Da Rold[3], Andrea Mattera[1], Gianluca Baldassarre[1]

**1** Laboratory of Computational Embodied Neuroscience, Institute of Cognitive Sciences and Technologies, National Research Council of Italy, Rome, Italy, **2** School of Computing, Electronics and Mathematics, University of Plymouth, Plymouth, United Kingdom, **3** Body Action Language Lab, Institute of Cognitive Sciences and Technologies, National Research Council of Italy, Rome, Italy

* giovanni.granato@istc.cnr.it

**Data Availability Statement:** All relevant data are within the manuscript and its Supporting information files.

**Funding:** GG: received funding from the European Union's Horizon 2020 Research and Innovation

## Abstract

Categorical perception identifies a tuning of human perceptual systems that can occur during the execution of a categorisation task. Despite the fact that experimental studies and computational models suggest that this tuning is influenced by task-independent effects (e.g., based on Hebbian and unsupervised learning, UL) and task-dependent effects (e.g., based on reward signals and reinforcement learning, RL), no model studies the UL/RL interaction during the emergence of categorical perception. Here we have investigated the effects of this interaction, proposing a system-level neuro-inspired computational architecture in which a perceptual component integrates UL and RL processes. The model has been tested with a categorisation task and the results show that a balanced mix of unsupervised and reinforcement learning leads to the emergence of a suitable categorical perception and the best performance in the task. Indeed, an excessive unsupervised learning contribution tends to not identify task-relevant features while an excessive reinforcement learning contribution tends to initially learn slowly and then to reach sub-optimal performance. These results are consistent with the experimental evidence regarding categorical activations of extrastriate cortices in healthy conditions. Finally, the results produced by the two extreme cases of our model can explain the existence of several factors that may lead to sensory alterations in autistic people.

## 1 Introduction

Human cognition evolved several perceptual mechanisms for adapting itself to categorise the surrounding world. Despite the fact that many brain structures innately code specific physical regularities of the world, there are learning mechanisms that allow the adaptation of perceptual processes to the environment demands. For example, during the solution of a categorisation task sensory processes increase the between-category differences and decrease the within-category difference, a phenomenon called 'categorical perception' (CP; [1, 2]). In [3] we

Program, under Grant Agreement No 713010 of the project 'GOAL-Robots – Goal-based Open-ended Autonomous Learning Robots' FD: received funding from the H2020-MSCA-IF-2017, under Grant Agreement No 796135 of the project 'INTENSS'. The funders had no role in study design, data collection and analysis, decision to publish, or preparation of the manuscript.

**Competing interests:** The authors have declared that no competing interests exist.

corroborated the idea that, during category learning, a CP-like effect can be supported by a top-down selection of perceptual representations. In that work, we have assumed that perceptual learning processes have previously created the category-based representations, which the agent select during the task solution. Here we investigate how these learning processes lead to category-based sensory representations, i.e. CP.

Experimental evidence suggests that these learning processes can occur in a bottom-up way, depending on the experienced input patterns [4–6], and in a top-down way, depending on task-dependent feedback signals [7–9]. However, there is controversial evidence regarding visual stages that show a CP effect. For instance, [10] found that CP influences the early stages of sensory processing (e.g., V1) and [11] propose that later cognitive stages of processing support CP (e.g., linguistic labels). Moreover [12], empirically corroborated the idea that both striate and extrastriate cortices support CP. Reconciling controversial results [13], propose that perceptual learning processes occur at different stages of visual hierarchy, depending on the task demands. In addition, several studies [3, 14, 15] suggest that the subcortical structures that support reward-based feedback signals (e.g. basal ganglia) interact with cortical structures, contributing to the emergence of category-based perception. In particular, many studies (for an extended review see [14]) suggest that dopamine-based reinforcement learning signals could affect category-related activations in visual sensory cortices. On the basis of this evidence, we have recently proposed the *Superlearning hypothesis* [16] suggesting that different learning mechanisms, such as Hebbian unsupervised learning (UL; [17]), and reward-based reinforcement learning (RL; [18]), can contemporarily occur within the same structures. Overall, this evidence suggests that: (1) the emergence of CP in sensory cortices is supported by bottom-up unsupervised signals, and category-based activations are affected by feedback-based reinforcement learning mechanisms; (2) CP-based representations may occur at different stages of cortical visual hierarchy; (3) the emergence of CP is supported by cortical mechanisms and subcortical structures signals (e.g. reward).

Some computational models have recently investigated the learning mechanisms leading to categorical perception effects (see section 4.3). These models take different approaches, as they focus on the interaction between low-level and high-level information at different neuronal sites (e.g. apical and basal dendrites; [19]), systems supporting speech production [20], self-organising mechanisms [21], visual competitive hierarchies [22], and effects of supervised signals [23]. Other models of CP investigate Bayesian inferential mechanisms [24] and embodied evolutionary influences [25]. Although these models clarify many aspects of CP, none of them focuses on the computational effects caused by an *interaction* between unsupervised and reinforcement learning processes.

By integrating studies on categorical perception, brain learning processes, and the Superlearning hypothesis, we propose here a computational model to investigate the interaction effects of UL and RL occurring during the acquisition of categorical perception. In particular, we exploit machine learning techniques to build a system-level neuro-inspired architecture, integrating an actor-critic approach [18] with a generative neural network [26]. Despite we adopted ML techniques, our model shows both a neuro-inspired architecture (the model emulates the interactions of perceptual, motivational and motor brain systems) and 'bio-plausible' learning mechanisms (e.g. localistic learning rules and a distributed encoding of representations; [27]). This approach allows us to model the functional features of brain systems supporting human cognition (see the Methods section). Taking inspiration from experimental protocols in category learning studies [28], we have tested the model with a simple sorting task that requires performing consistent actions in response to one of three visual features (i.e. colour, shape, or size) of simple geometric images.

The results show that a balanced mix of UL and RL processes in the perceptual component leads to higher performance. In particular, the learned representations exhibit a categorical perception effect, for which the representations of inputs included in a specific category (e.g., red inputs) tend to be encoded with similar neural patterns and to differ from patterns encoding other categories (e.g., blue, green, and yellow groups). Interestingly, the representations developed by the UL/RL balanced model encompasses both intrinsic statistical regularities and action-relevant visual features of images. Instead, the models using only RL or UL exhibit a sub-optimal performance due to an impaired categorical perception effect, resulting in an inefficient, slow, and variable representation learning process.

Our computational analysis and results represent a source of information to explain experimental data and categorisation deficits in clinical conditions. For example, despite sensory alterations in autism are commonly attested [29–31], contrasting proposals suggest that in autism the categorisation skills are corrupted by a weak feedback integration [32] or an extreme reward-dependent learning [15]. Our model explains these different sub-optimal categorisation mechanisms with the emergence of different heterogeneous autism spectrum conditions. At last, our work represents a prompt for machine learning and robotics fields. For example, ML proposals start to integrate different learning rules to improve discrimination competences [33, 34] and to obtain more robust representations and categorical perception effects [35, 36]. Our neuro-inspired algorithm could represent a starting point to build efficient algorithms that balance UL and RL depending on the task demands and generalisation requirements.

## 2 Methods

This section presents the description of the task and of the computational model. Note that, as previously done [3, 37], a neuroscientific and theoretical investigation of brain systems supporting the investigated phenomenon (categorical perception) preceded and guided the model building. Therefore, in section 2.2 we describe the theoretical and neural underpinnings of the model functioning, while in section 2.3 we describe the computational algorithms and artificial neural networks that support the model components.

### 2.1 Task and experimental conditions

The experimental protocol is composed of a 'pre-task section' and a 'task performance section' (Fig 1A). In the pre-task section, the environment chooses a specific sorting rule (i.e. colour, shape, or size) and creates a set of 'ideal vectors'. These vectors correspond to the output vectors that the model should produce in correspondence to a specific input and a specific sorting rule. In this way, in each trial a visual input is provided to the model and the environment computes feedback (reward) on the basis of the distance between the model response and the ideal response (see Section 2.3 for further details). For example, in case the environment chooses 'colour' as a sorting rule, all inputs with a specific colour (i.e. red, green, blue, or yellow) will be associated with one of four ideal vectors. The second section of the protocol is composed of many trials. Within each trial, the model interacts with a virtual environment through four phases (Fig 1A, on top). First, the environment provides a single visual input to the model that processes it (phase 1). The visual input is extracted from a set of 2D input images of geometrical shapes varying in colour, shape, and size, produced from four example images (Fig 1B). Second, the model produces an output (distributed binary vector) on the basis of the processed visual input (phase 2). Third, the environment returns a score index that suggests the correctness of the model response with respect to the ideal one (phase 3). Fourth, the model computes the reward returned by the environment and adapts its internal

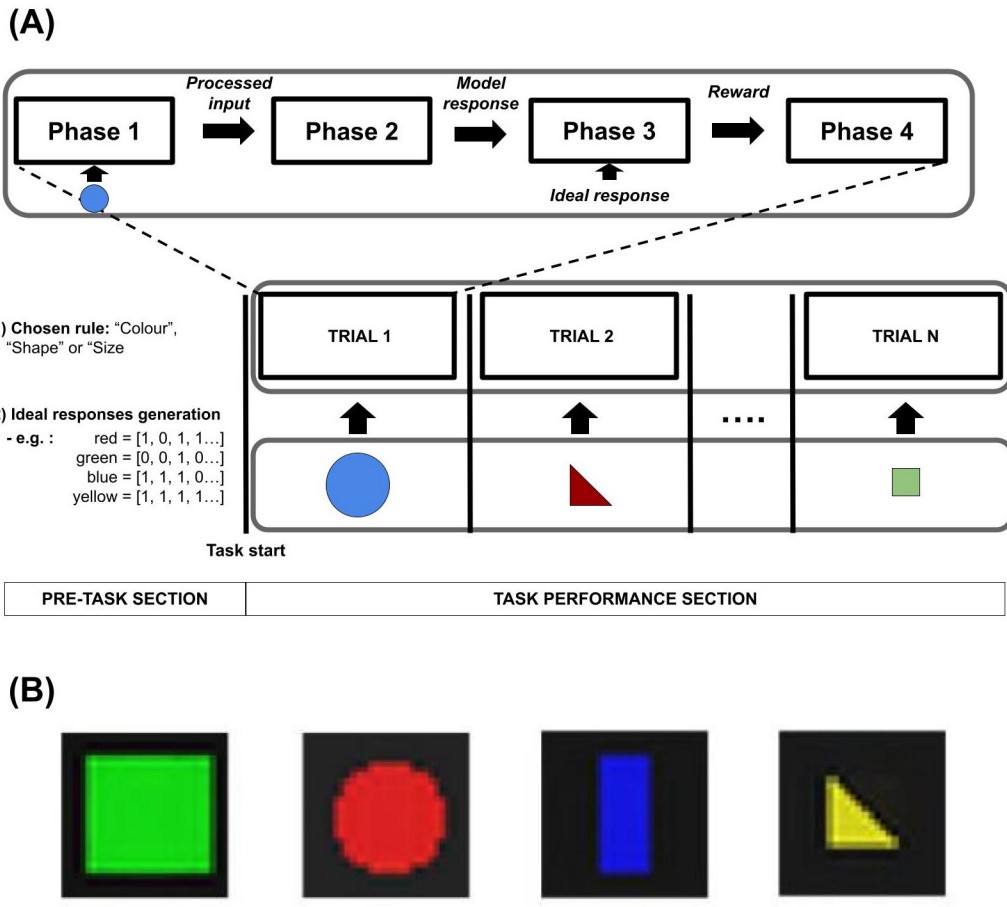

**Fig 1.** (A): Scheme of the task protocol. The row below shows the examples of inputs that the environment provides to the model (visual input). The middle row shows the trial sequence. Note that a first experimental section occurs before the task experimental section with trials and involves the setting of the task conditions, that is, the choice of the sorting rule and the creation of the ideal responses. The top row zooms in a specific trial, showing the phases that occur during the model-environment interactions. (B): Examples of the 64 geometrical shapes (circles, squares, rectangles, triangles), used to produce the images. Each image encompasses a different attribute out of the four attributes of each of the three categories, namely shape, colour, and size.

components (phase 4). The set of trials (64 stimuli) is repeated a certain number of times in random order.

Overall, the task we used to test the model is inspired by category learning tasks, requiring the production of a response on the basis of specific visual features of stimuli such as colour, shape, and size (see [38, 39] for an extended analysis of these tasks).

In particular, we focused on a subclass of these tasks in which a classification rule is fixed and the participant has to execute a motor action on the basis of the features of a card [28]. Note that despite the task being inspired by experimental protocols, the same learning processes we emulate could support the ecological development of infants' categorical perception [40, 41].

## 2.2 Neuro-inspired underpinnings of the model: Main learning processes and key components

Fig 2A summarises the main 'model-environment interactions' during the task performance: perception of the input (bottom-up spread of input information from the world), behavioural

**Fig 2.** (A) A schema of the main model processes involved in its interaction with the environment during the task performance. (B) Scheme of learning processes and targeted brain areas that are addressed by the hypothesis and computational model presented here. The intermediate sensory-motor layers (extrastriate cortices) undergo both associative unsupervised learning (UL) and trial-and-error learning (RL). The latter presents a gradient having a decreasing strength moving from the motor cortex towards the striate cortex.

response (production of an output toward the world), feedback computation (computation of the external world feedback, e.g. reward signal), and learning (reward-based adaptation of sensory-motor processes). The top of the figure highlights a 'sensory-motor loop', in which the model iteratively perceives the world and executes an action. The bottom of the figure highlights a 'learning loop', for which the model adapts its sensory computation, behaviour and feedback computation through a learning process.

Albeit in a simplified form, the learning processes of the model and its loops with the environment are coherent with the theoretical framework of *embodied perception*. Indeed, many studies [42–45] propose that the brain constructs internal representations of the world 'for being ready to act', also establishing a relation between embodied cognition and categorical perception [46–48]. Although the model does not show a full embodiment due to the lack of specific actuators with realistic physical dynamics (see section 4.4 for an analysis of this limitation and a possible solution), it has some features that move towards embodiment. In particular, the world affects the perceptual states of the agent, in turn affecting its response; then categorical perception emerges as a consequence of the feedback provided by the world. According to some views, [45], these elements of the model-environment interactions represent a key feature of embodiment.

The architecture and learning processes of the model are inspired by the interactions between brain cortical and subcortical macro-systems (e.g. striate and extrastriate cortices, basal ganglia, motor cortices) that support the computational functions we investigate here (e.g. perceptual abstraction, motivational bias, motor selection). For example, fMRI experiments on humans and monkeys show that most cortical regions are activated by reward signals with a trial-locked timing [49–52] and the dopamine probably mediates these reward-related signals [50, 52]. Furthermore, evidence from discrimination tasks [49, 50] suggests that the reward-induced reactivation of the sensory cortex tunes the representations in a task-dependent way. At last, there is experimental evidence (for an extended review see [14]) suggesting that the dopamine-based RL signals could affect the categorisation processes in sensory cortices, of which CP is an instance. Integrating this evidence, the Super-learning theory [16] proposes that different learning signals can coexist in the same brain structure (e.g. associative Hebbian and reward-based mechanisms).

On the basis of this experimental evidence and studies on categorical perception, our architecture integrates two features that are shown in Fig 2B. First, in sensory-motor hierarchy intermediate layers (extrastriate cortices) host mixed UL and RL processes while early layers (striate cortex) and later layers (motor cortices) respectively host unsupervised and

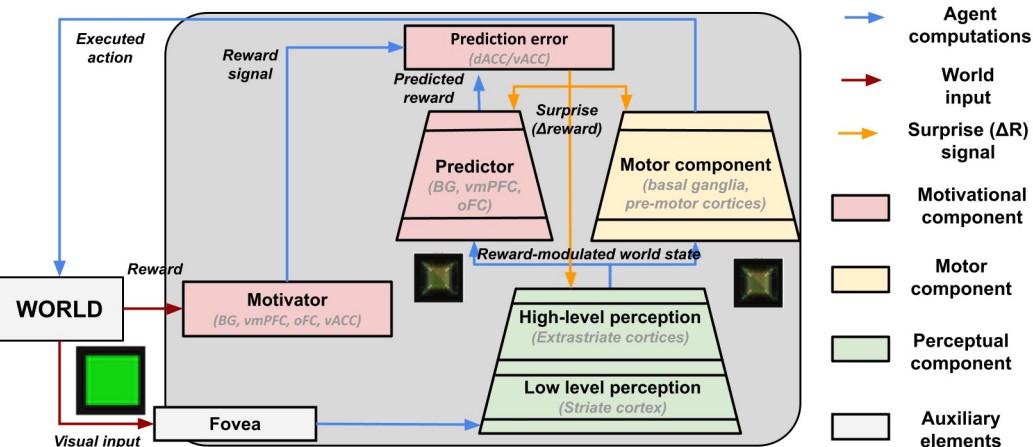

**Fig 3. Schema of the model components and functions, the flows of information between the components, and the learning signals.**

reinforcement learning mechanisms. Second, task-dependent signals from the world (i.e. rewards) direct reach the perceptual component, without adopting an implausible error back-propagation mechanism [27]. This proposal represents a simplified solution but it captures the macro differences in learning processes that could lead to categorical perception.

Here we specifically focus on categorical perception and its possible relationship with the existence of UL/RL interactions suggested by the Superlearning hypothesis [16]. However, our solution is only one possible approach to the investigation of the brain's adaptive learning dynamics. Indeed, alternative views propose higher segregation of the learning modalities in the brain [53] and other modelling approaches emulate the emergence of adaptive dynamics adopting a pure UL approach [54] or pure RL approach (e.g. meta-RL; [55]).

As in [3], the model abstracts the fine-grain biological details (e.g. neuronal micro-circuitry or bio-grounded plasticity). However, the interactions between the macro-systems underpinning the learning processes (e.g. motivational and perceptual systems interactions) are bio-plausible (e.g. localistic learning rule and distributed representations coding; [27]). This level of detail is suitable for investigating the computational mechanisms that support the human learning processes underlying categorical perception.

Fig 3 shows the whole model architecture and the information flows between its components, also reporting the brain structures from which the components are inspired. Despite the model showing some simplifications, it proposes a system-level architecture that represents a promising approach in the computational modelling field [56]. The functional neural underpinning of the model components and learning processes are now explained in-depth, while implementation details are reported in section 2.3.

**2.2.1 Perceptual component.** This component is based on a neural network that receives visual inputs and performs information abstraction, mimicking the brain visual system. In particular, the component emulates hierarchical information processing [57, 58] from the low-level retinotopic features in striate cortex to the high-level features (e.g. colour, shape, size) in extrastriate cortices [59, 60].

Differently, from the biologically implausible gradient-descent methods, the network learns through a bio-plausible mechanism [27]. In particular, the learning rules update each connection weight (synapse) on the basis of locally available information related to the pre-synaptic and post-synaptic units. The distributional coding of representations is another biologically

plausible feature of the model. Indeed, information on each content (e.g., a percept) is encoded by many units of the layer, and each unit takes part in the representations of different contents. This encoding is more bio-plausible than localistic representations ('grandmother cells; [61, 62]). Finally, the differences in learning processes of the model layers represent a further bio-plausible feature. In particular, the top layer of this component, emulating extrastriate cortices, is trained through a mechanism that integrates associative and reward-based RL (Fig 2B). Instead, the bottom layer of the component, which mimics early visual cortices, is trained before the task execution reflecting an early development [63]. Critical for our hypothesis, these features capture the essence of the different weights that reward signals (e.g. dopamine-based inputs) have onto extra-striate and striate cortices [64–68].

**2.2.2 Motor component.** This component is supported by a neural network that, on the basis of the perceptual component activation, produces an 'action' affecting the world. The network is trained through a trial-and-error learning algorithm using a reward signal, mimicking the interactions of basal ganglia with motor cortices during the learning of actions [69, 70].

**2.2.3 Motivational component.** This component is formed by three sub-modules that emulate the motivational functions supported by different brain sub-systems.

First, a *motivator* sub-module produces a reward signal on the basis of the action outcome. Here the outcome is received from the environment and informs the system on the 'correctness' of the performed action (see below). This action-outcome might correspond to an 'extrinsic reward' (e.g. food or other rewarding resources) and is suitably processed by the system sensors and motivator component to produce a reward signal. Alternatively, the reward signal might be produced by intrinsic motivation processes [71, 72] related to the novelty or surprise of the experienced stimuli [73] or to the goal-directed acquisition of competence [74, 75]. In the brain, subcortical and ventral cortical structures support extrinsic rewards [76, 77] while other subcortical and dorsal cortical structures support the computation of intrinsic reward signals [72, 78, 79].

Second, a *predictor* sub-module, based on a multi-layer neural network, uses the representations of the top layer of the perceptual component to predict future rewards. This module functionally mimics the brain basal-ganglia striosomes [80].

Last, a *prediction error* sub-module integrates the obtained and predicted rewards and produces a learning signal ('surprise'). This signal influences the learning of the predictor, of the motor component and, most importantly, of the perceptual component. In the brain, this signal is represented by the phasic dopamine bursts reaching various target areas [81], and it has been modelled by the actor-critic RL architecture [82].

## 2.3 Computational implementation and learning algorithms of the model

The architecture (Fig 4) is formed by a generative model integrated into an actor-critic architecture [83], both modified to study the role of unsupervised and reinforcement learning supporting the emergence of categorical perception. Moreover, auxiliary computational elements support the interaction between the model and an abstract task protocol (e.g. the world feedback).

A global view of the model networks and further details regarding the system parameters (e.g., the number of units of each layer, the learning rates, the training epochs, etc.) are reported in S1 Fig and Table 1 of S1 text in S1 File. The code of the system will be made publicly available online on GitHub in case of publication.

**2.3.1 Perceptual component.** This component is a generative *Deep Belief Network* (DBN; [84, 85]) composed of two stacked *Restricted Boltzmann Machines* (RBM; [86]). Each RBM is composed of an input layer ('visible layer') and a second layer ('hidden layer') formed by

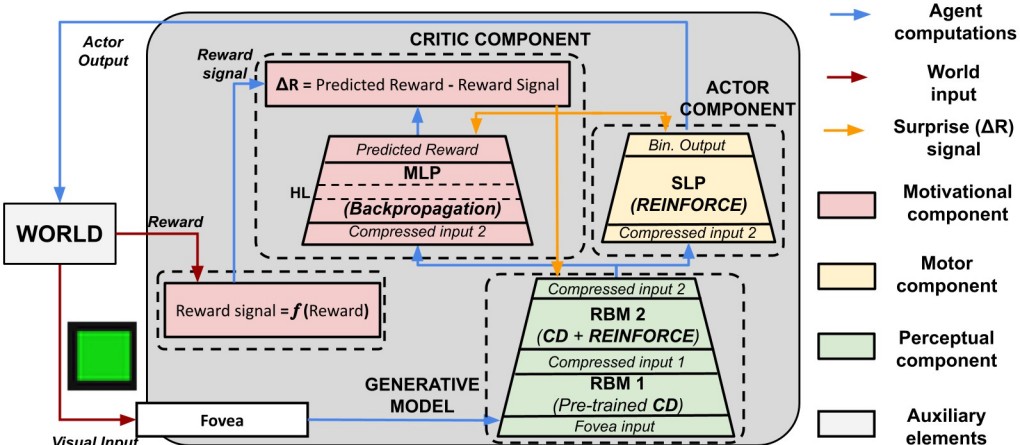

**Fig 4. A computational schema of the model components and their training algorithms, the flows of information between the components, and the learning signals.** MLP: Multi-layer Perceptron. SLP: Single-layer Perceptron. HL: Hidden Layer. RBM: Restricted Boltzmann Machine. CD: Contrastive Divergence.

Bernoulli-logistic stochastic units where each unit $j$ has an activation $h_j \in \{0, 1\}$:

$$h_j = \begin{cases} 1 \; \; if \; \; v \geq \sigma(p_j) \\ 0 \; \; if \; \; v < \sigma(p_j) \end{cases} \tag{1}$$

$$\sigma(p_j) = \frac{1}{1 + e^{-p_j}}$$

$$p_j = \sum_i (w_{ji} \cdot v_i)$$

where $\sigma(x)$ is the sigmoid function, $p_j$ is the activation potential of the unit $h_j$, $v$ is a random number uniformly drawn from (0, 1) for each unit, and $w_{ji}$ is the connection weight between the visible unit $v_i$ and $h_j$. The RBM is capable of reconstructing the input by following an inverse activation from the hidden layer to the input layer.

The DBN consists of a stack of RBMs—two in the model—where each RBM receives as input the activation of the hidden latent layer of the previous RBM. The model is trained layer-wise, starting from the RBM which receives inputs from the environment and towards the inner layers. On this basis, the DBN executes an incremental dimensionality reduction of the input, as higher layers further compress the representations received from the lower/previous RBM [87]. In the model, the first RBM directly receives the input images and it is trained to encode them 'offline' before the task. This training adopts the *Contrastive Divergence* (CD), an unsupervised learning algorithm that computes each connection weight update $\Delta w_{ij}$ on the basis of a bidirectional iterative process (see S2 Fig in S1 text in S1 File—for a graphical representation of the RBM training with CD). In particular, the visible layer receives an external input and activates the hidden layer, which in turn re-activates the previous visible layer (the weights of an RBM are bidirectional). Then, this reactivated visible layer activates the hidden layer for the second time. This cycle, involving a direct and inverse spread of the input, can be repeated many times but it is usually performed two times (visible-hidden-visible-hidden activation). The first cycle of visible-hidden layer activations are usually labelled as 'data'

activations, in that are directly caused by the external data (input). Differently, the second cycle of visible-hidden layer activations is usually labelled as 'model activations' or 'reconstructions'. The following formula describes the CD algorithm:

$$\Delta w_{ij} = \epsilon(\langle v_i \cdot h_j \rangle_{data} - \langle v_i \cdot h_j \rangle_{model}) \qquad (2)$$

where $\epsilon$ is the learning rate, $\langle v_i \cdot h_j \rangle_{data}$ is the product between the initial input (initial visible activation) and the consequent hidden activation, $\langle v_i \cdot h_j \rangle_{model}$ is the product between the reconstructed visible activation and a second activation of the hidden layer following it, averaged over all data points.

The second RBM of the model is trained 'online' during the task performance based on the novel algorithm proposed here. The algorithm integrates *Contrastive Divergence* (Eq 2) with the *REINFORCE* algorithm described in the next session (Eq 4) as follows:

$$\Delta w_{ij} = \quad \lambda \ (\epsilon \ (\langle v_i \cdot h_j \rangle_{data} - \langle v_i \cdot h_j \rangle_{model})) \ +$$
$$(1 - \lambda) \ (\alpha \ (r - \bar{r})(y_j - p_j)x_i) \qquad (3)$$

where $\lambda$ is the contribution of Contrastive Divergence to the update of weights, and $(1 - \lambda)$ the contribution of REINFORCE. Crucial for this work, $\lambda$ mixes the contribution of UL and RL processes to the weight update, in particular, a high value implies a dominance of UL whereas a low value implies a dominance of RL. In the simulations, we tested five values of the parameter: $\lambda \in \{1, 0.1, 0.01, 0.001, 0\}$.

**2.3.2 Motor component.** This component is a single-layer perceptron trained with the RL algorithm REINFORCE [88]. The input of the network is the activation of the last layer of the perceptual component. The network output layer is composed of Bernoulli-logistic units as for the perceptual component. The algorithm computes the update $\Delta w_{ji}$ of each connection weight linking the input unit $i$ and the output unit $j$ of the component as follows:

$$\Delta w_{ji} = \alpha(r - \bar{r})(y_j - \sigma(p_j))x_i \qquad (4)$$

where $\alpha$ is the learning rate, $r$ is the reward signal received from the motivator, $\bar{r}$ is the reward signal expected by the predictor, $x_i$ is the input of the network (from the outer second hidden layer of the DBN), $\sigma(p_j)$ is the sigmoidal activation potential of the unit encoding its probability of firing, and $y_i$ is the unit binary activation.

**2.3.3 Motivational component.** This component implements the functions of the *critic* component of an *actor-critic* architecture [83].

The *motivator* module computes the reward signal by scaling the reward perceived from the external environment into a standard value, the *reward signal* $r \in (0, 1)$:

$$r = f(Reward) \qquad (5)$$

where *Reward* is the reward perceived from the environment and $f(.)$ is a linear scaling function ensuring that the reward signal ranges between 0, corresponding to a wrong action, to 1, corresponding to an optimal action. This reward signal represents the pivotal guidance of the RL processes. As discussed in the previous sub-section, in other cases the motivator may involve further mechanisms, computing the reward signals on the basis of extrinsic and/or intrinsic motivation mechanisms.

The *predictor* module is a multi-layer perceptron composed of an input layer, a hidden layer, and an output layer. The input layer corresponds to the second hidden layer of the DBN while the output layer, composed of a single linear unit, corresponds to the expected reward signal $\bar{r}$ computed on the basis of the DBN activation. The perceptron is trained with a

standard gradient descent method [61, 89] using a learning rate $\alpha$ and the error $e$ computed by the prediction-error component.

The *prediction error* module is a function that computes the reward prediction error (surprise) $e$ as follows:

$$e = r - \bar{r} \tag{6}$$

where $r$ is the reward signal from the motivator, and $\bar{r}$ is the expected reward signal produced by the evaluator. This error is used to train the predictor itself, the motor component, and the perceptual component.

**2.3.4 Auxiliary elements.** The input dataset is formed by RGB images with a black background and a polygon at the centre (Fig 1B). The polygon is characterised by a unique combination of specific attributes chosen from three visual categories: colour, form, and size. There are four attributes for each category: red, green, blue, yellow (colour); square, circle, triangle, bar (shape); large, medium-large, medium-small, small (size). These attributes generate $4^3 =$ 64 combinations forming the images used in the test.

The retina component is implemented as a $28 \times 28 \times 3$ matrix containing the RGB visual input. The matrix is unrolled into a vector of 2, 352 elements that represents the input of the perceptual component.

The environment (1) chooses the correct sorting rule before the task performance and creates a set of ideal actions for each input, and (2) provides an image to the model at each trial. In every trial, the model perceives and processes one input image (Fig 1A) and undergoes a cycle of the aforementioned learning processes based on the reward received from the environment after the action performance (Fig 2A). Here the environment computes the reward $r'$ simply on the basis of the Euclidean distance between the model action and an 'optimal action':

$$Reward = \|\mathbf{y}^* - \mathbf{y}\|_1 \tag{7}$$

where $\mathbf{y}^*$ is the optimal action binary vector that the model should produce for the current input, $\mathbf{y}$ is the model binary action, and $\|.\|_1$ is the L1 norm of the vector difference. The optimal actions are four binary random vectors that the model should produce in correspondence to the items of the four input categories of the given task.

## 3 Results

We tested the model with different task conditions and model configurations. First, we varied the sorting rule, hence the task shows three task conditions. For example, a specific task condition required sorting the cards by colour and another one by shape or by size. The sorting rule is fixed before the task starts and it does not change during the task performance. Second, we tested the model with five different levels of UL/RL contribution ($\lambda$ parameter, see Section 2.3). This variation gave rise to five model conditions, labelled as follows: Level 0 (L0): no RL (i.e., only UL); Level 1 (L1): low RL; Level 2 (L2): moderate RL; Level 3 (L3): high RL; Level 4 (L4): extreme RL (no UL). Third, we tested the model with two further conditions, namely 10 and 50 units in the second DBN hidden layer. These conditions aim to test the impact of the available computational resources (i.e. the number of suitable units for storing the input information) on the task performances.

We varied the parameters of these environmental and model conditions with a random grid search based on over 1000 simulations. The simulations were run in the *Neuroscience Gateway platform* [90].

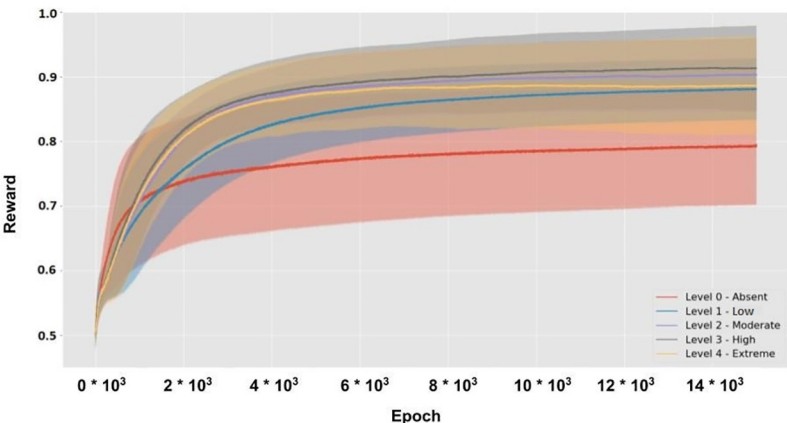

**Fig 5. Reward per epoch of the five models involving different UL/RL levels, averaged over the models using a given level.** Shaded areas represent the standard deviation.

The presentation of results is organised in three parts. The first part investigates the relationship between the specific UL/RL balances and the task performance. The second part investigates the relationship between the specific UL/RL balances and the nature of the perceptual representations acquired. Finally, the third part presents a graphical visualisation of the previous representations and an analysis of the amount of information (visual details) they stored.

## 3.1 Performances analysis

Fig 5 shows the training curves of the models, trained with different RL contributions in 15,000 epochs. The L0 models, using only UL, learn faster during the first 1,000 epochs but exhibit the worst final performance. S3–S5 Figs in S1 text in S1 File show that this effect is present in most of the simulations. Instead, the highest final performance is achieved by the L3 and L2 models where UL and RL are better balanced. Fig 6 shows the final performance of the models, namely the maximum reward they achieved.

A one-way ANOVA confirms the presence of a statistical difference between the final performance of the five groups ($F = 47.51$, $p < 0.001$). Post hoc tests (Table 1) confirm that the performances of models with an absent RL contribution (L0) are statistically different with respect to each of the other models ($0.81 \pm 0.08$, all $p < 0.001$). The L3 models show a higher performance compared to the L0 models ($0.92 \pm 0.06$ vs. $0.81 \pm 0.08$, $p < 0.001$), the L1 models

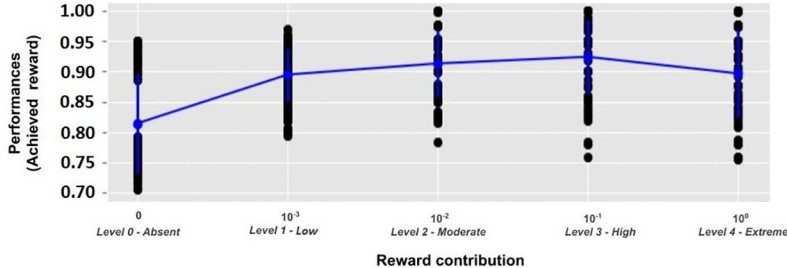

**Fig 6. Performances (maximum reward obtained at the end of training) of models featuring different levels of RL contribution.**

**Table 1. Post-hoc comparisons (t-test with Bonferroni correction) between the performance of models with different levels of RL contribution.**

|  | Absent (L0) | Low (L1) | Moderate (L2) | High (L3) | Extreme (L4) |
|---|---|---|---|---|---|
| **Absent (L0)** | // | // | // | // | // |
| **Low (L1)** | $p < 0.001$ | // | // | // | // |
| **Moderate (L2)** | $p < 0.001$ | $p > 0.05$ (NS) | // | // | // |
| **High (L3)** | $p < 0.001$ | $p < 0.001$ | $p > 0.05$ (NS) | // | // |
| **Extreme (L4)** | $p < 0.001$ | $p > 0.05$ (NS) | $p > 0.05$ (NS) | $p < 0.05$ | // |

'NS' indicates 'non statistically significant'.

($0.92 \pm 0.06$ vs. $0.89 \pm 0.04$, $p < 0.001$), and the L4 models ($0.92 \pm 0.06$ vs. $0.90 \pm 0.07$, $p < 0.05$). The L2 and L3 models do not show a significant difference ($0.9 \pm 0.06$ vs. $0.91 \pm 0.05$).

To further investigate the relationship between the performance of the models and the different levels of RL contribution, we grouped the results of the simulations on the basis of the computational resources or the sorting rule (Table 2). Here we present a summary of the results while Section 2.1 in the S1 text in S1 File reports the posthoc tests.

Overall, increasing available computational resources tends to lower the amount of RL contribution needed to achieve the highest performance. Indeed, a one-way ANOVA shows a statistical difference between the models ($F = 3.85$, $p < 0.001$) and the post-hoc tests show that the L2 model leads to the best result ($0.95 \pm 0.05$).

The table also highlights differences between the simulations using different sorting rules (colour, shape, size). The simulations with the *colour sorting rule* show flattened reward values with respect to the different RL contributions. In the case of low computational resources the model does not show statistically significant differences ($F = 0.88$, $p > 0.05$). A difference emerges in the case of high computational resources ($F = 19.8$, $p < 0.001$) where the L2 models, having a balanced UL/RL mix, show the best final performance ($0.98 \pm 0.02$).

The simulations with the *shape sorting rule* show statistical differences with both low computational resources ($F = 120.9$, $p < 0.001$) and high computational resources ($F = 20.4$, $p < 0.001$). In both cases, the models using a mixed level of UL and RL prevail: the extreme cases of the L0 models (only UL), and L4 models (only RL) have lower performances with respect to the L1, L2 and L3 models having a more balanced UL/RL mix.

**Table 2. Performance of models with different RL contributions in correspondence to two different amounts of computational resources (number of neurons in the second hidden layer of the DBN) and three different sorting rules (colour, shape, size).**

|  | Absent | Low | Moderate | High | Extreme |
|---|---|---|---|---|---|
| **Low Resources (Average)** | $0.81 \pm 0.08$ | $0.89 \pm 0.04$ | $0.91 \pm 0.05$ | **0.92** $\pm 0.06$ | $0.90 \pm 0.07$ |
| Colour | $0.92 \pm 0.02$ | **0.92** $\pm 0.02$ | $0.91 \pm 0.04$ | $0.91 \pm 0.07$ | $0.90 \pm 0.08$ |
| Shape | $0.75 \pm 0.02$ | $0.89 \pm 0.04$ | $0.94 \pm 0.04$ | **0.95** $\pm 0.04$ | $0.93 \pm 0.06$ |
| Size | $0.76 \pm 0.02$ | $0.88 \pm 0.05$ | $0.89 \pm 0.06$ | **0.90** $\pm 0.06$ | $0.86 \pm 0.07$ |
| **High Resources (Average)** | $0.92 \pm 0.03$ | $0.93 \pm 0.04$ | **0.95** $\pm 0.05$ | $0.93 \pm 0.06$ | $0.93 \pm 0.05$ |
| Colour | $0.94 \pm 0.01$ | $0.94 \pm 0.01$ | **0.98** $\pm 0.02$ | $0.95 \pm 0.03$ | $0.96 \pm 0.02$ |
| Shape | $0.93 \pm 0.02$ | $0.97 \pm 0.02$ | **0.97** $\pm 0.02$ | $0.96 \pm 0.02$ | $0.94 \pm 0.02$ |
| Size | $0.88 \pm 0.02$ | $0.88 \pm 0.03$ | **0.90** $\pm 0.05$ | $0.88 \pm 0.07$ | $0.88 \pm 0.07$ |

Labels with '(Average)' identify the average of the three conditions (colour, shape, size) in case of low or high resources. Values in bold highlight the highest value for each condition (along the rows).

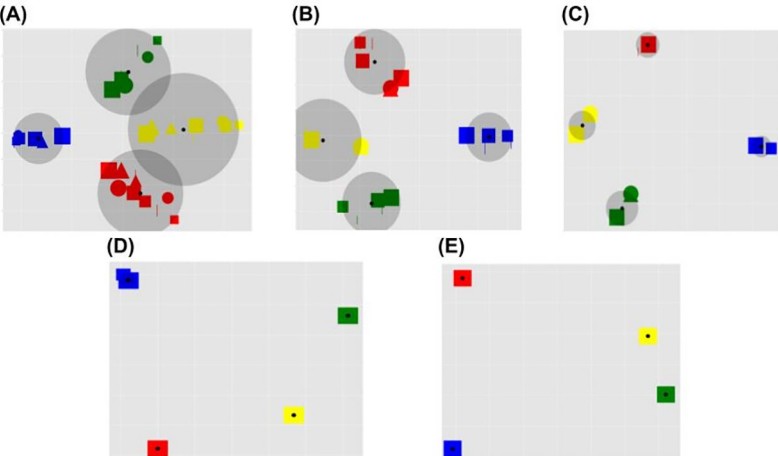

**Fig 7. Colour sorting category: Reconstructed input.** Principal components of the reconstructed image representations in the case of the colour sorting rule and in correspondence to different levels of RL (shown in different graphs). The dimensionality of the reconstructed image was reduced to two through a PCA (x-axis: first component; y-axis: second component). Within each graph, each reconstructed image is represented by a point marked by an icon that summarises the colour, shape, and size of the shape in the image (some icons are not visible as they overlap). The centroids of the four clusters found by the K-means algorithm are marked with a black dot, while the maximum distance of the points of the cluster from its centroid is shown by a grey circle. A: Level 0 (L0), absent RL (only UL); B: Level 1 (L1), low RL; C: Level 2 (L2), moderate RL; D: Level 3 (L3), high RL; E: Level 4 (L4), extreme RL (no UL).

Finally, the simulations with the *size sorting rule* show statistical differences with low computational resources ($F = 43.4$, $p < 0.001$) but not with 'high computational resources' ($F = 1.12$, $p > 0.05$). In the first case, the L0 models have the lowest performance.

## 3.2 Analysis of internal representations

To investigate the nature of the perceptual representations acquired by the models, we show the results of some example simulations with different sorting rules and different levels of the RL (other simulations lead to qualitatively similar results). Since we have adopted 'realistic inputs' (geometric figures), we have analysed the 'reconstructed representations' of the input layer rather than the hidden representations (for a detailed description of this reconstruction procedure see section 2.2 in S1 text in S1 File). We adopt this strategy to better interpret the acquired representations of the original inputs, of which we can plot the original geometric images (see Fig 11). Our sample tests on the representations in the hidden layer show similar results.

To plot the representations we used a Principal Component Analysis (PCA), allowing a dimensionality reduction, and a K-means algorithm, supporting clustering. First, we extracted the first two principal components of the visible layer in correspondence to the original 64 input patterns. Second, The K-means algorithm was applied to the PCA results by setting $K = 4$, so that the algorithm grouped the representations into four classes, as the number of the actions.

Further details and results regarding these methods are reported in Section 2.3 of S1 text in S1 File.

The results (Figs 7–9) highlight that the RL contribution strongly affects the internal representations as revealed by the reconstructed inputs. Models with a medium (L2) and high (L3)

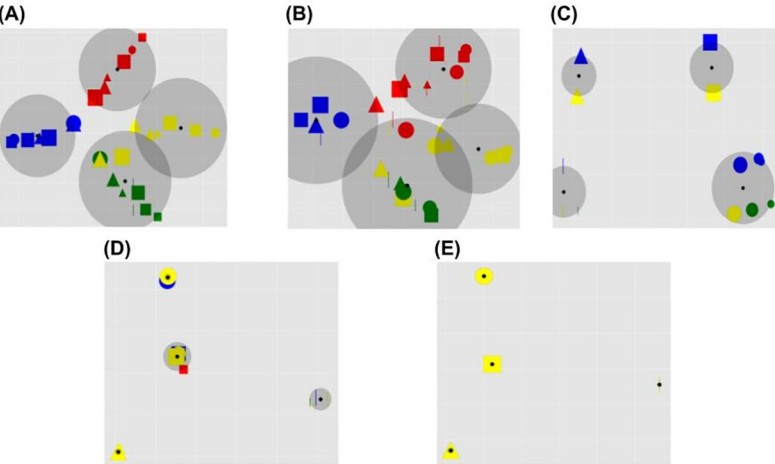

**Fig 8. Shape sorting category: Reconstructed input.** Principal components of the reconstructed image representations in the case of the shape sorting rule and in correspondence to different levels of RL. Note that, in case of overlap, the yellow inputs appear at the top and hide others due to technical factors (we plot the yellow inputs at the end). The plots are drawn as in Fig 7.

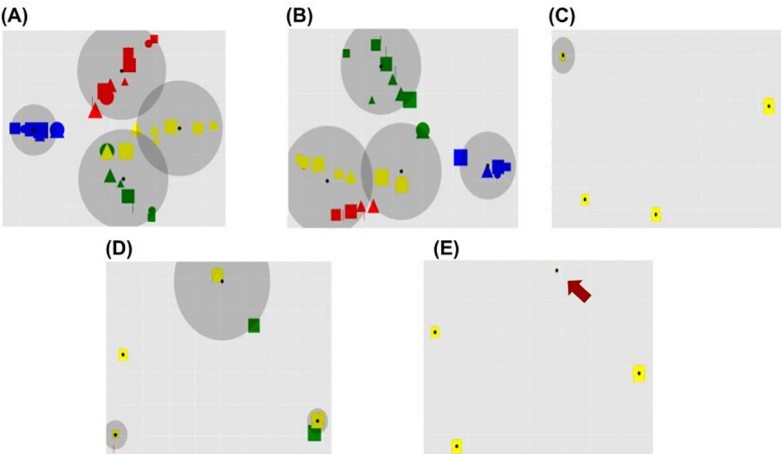

**Fig 9. Size sorting category: Reconstructed input.** Principal components of the reconstructed image representations in the case of the size sorting rule and in correspondence to different levels of RL. Note that, in case of overlap, the yellow inputs appear at the top and hide others due to technical factors (we plot the yellow inputs at the end). The graphs are drawn as in Fig 7. The red arrow in graph E indicates the centroid of a cluster that contains only the small bars but not the other small shapes.

level of RL show the emergence of task category-based clusters, whose radius progressively decreases as the weight of the RL increases. Conversely, the L0 and L1 models, with an absent or low RL, show a task-independent clustering effect on the basis of the input colours.

Fig 9E shows that the model with an extreme RL incurred in a clustering error. In particular, in this condition the model should group the images into four clusters (as in the conditions of Fig 9C and 9D) whereas it tends to use only three clusters and the fourth cluster on the right is almost empty.

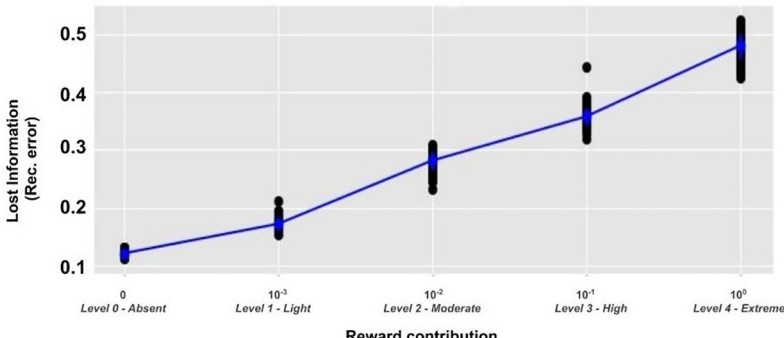

**Fig 10. Information loss for different levels of RL.** Information loss (reconstruction error at the end of the training) of models with different levels of RL.

## 3.3 Information stored by the model

To further investigate what type of information is stored by the perceptual representations, we show the results of two additional analyses. The first analysis examined the DBN reconstruction error (see Section 2.3 in S1 text in S1 File—for further details), while the second analysis qualitatively inspected the reconstructions of the input images.

Fig 10 shows the results of the first analysis and highlights the presence of a strong positive linear relationship between the level of RL and the reconstruction error ($r = 0.68$, $p < 0.001$).

A one-way ANOVA confirmed the presence of a statistical difference between the five groups ($F > 100.0$, $p < 0.001$). These results indicate that an increasing RL contribution causes a progressive loss of information on the input images.

The qualitative inspection of the reconstructions shows the kind of information that the internal representations tend to retain, in particular if the system tends to store task-independent and/or task-related features. In this respect, Fig 11 highlights the emergence of categorical perception, i.e. shapeless coloured blobs in case of colour sorting rule, colourless and sizeless prototypical shapes in case of shape sorting rule, and colourless blobs with different sizes in case of size sorting rule.

## 4 Discussion

### 4.1 Interpretation of the results

Here we discuss the results regarding the relationship between UL/RL contributions, behavioural performance and perceptual representations.

**4.1.1 Unsupervised learning, reinforcement learning and categorisation performances.** A main result of this work is that a suitable balanced mix of UL and RL leads the model to achieve the best performance in all tested conditions (Fig 6 and Table 1). Moreover, different UL/RL balances lead to different learning trends and behaviours of the models (Fig 5). For example, during the initial training phase, the model with an absent reward contribution (L0) has some advantages, exhibiting the sharpest increasing learning curve with respect to the models with a higher RL (L2, L3 and L4). S3–S5 Figs in S1 text in S1 File corroborate these results, showing that L0 (only UL) and L1 (low RL) have a learning advantage at early stages.

Functional analysis of the models with a higher RL can explain this effect. These models initially produce a slow and highly variable exploratory behaviour, resulting in more early unstable perceptual representations. The early slowness and variability are caused by the key

## Input reconstructions (sorting category: colour)

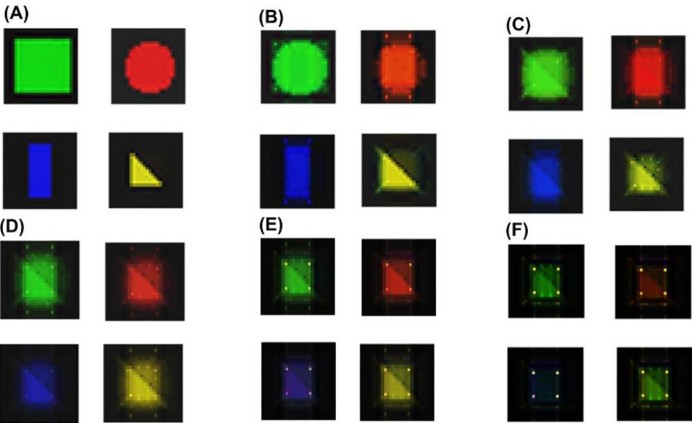

## Input reconstructions (sorting category: shape)

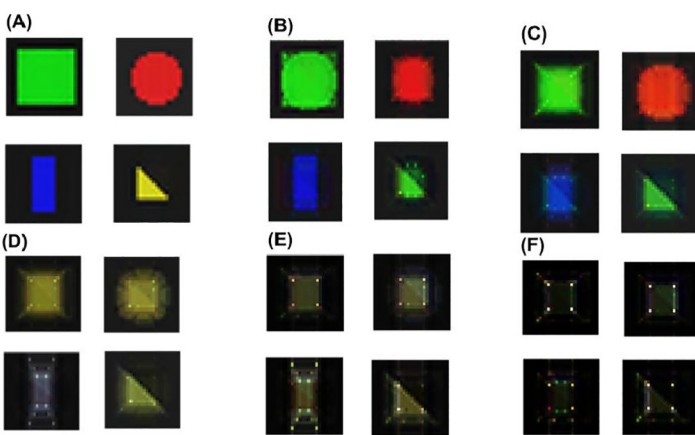

## Input reconstructions (sorting category: size)

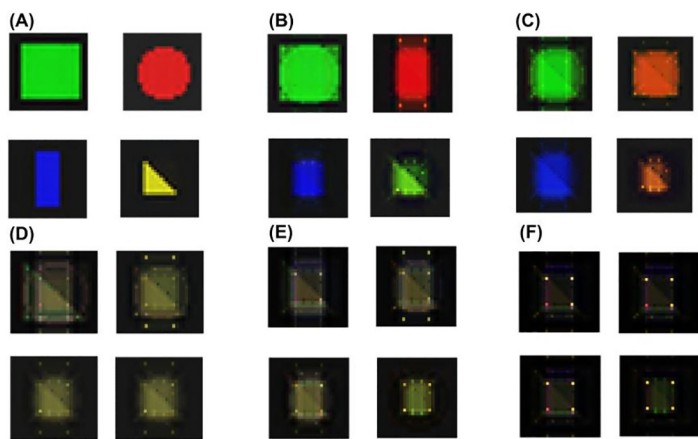

**Fig 11. Input reconstructions (sorting category: Colour) Input reconstructions (sorting category: Shape) Input reconstructions (sorting category: Size).** Image reconstructions with different sorting rules and different levels of RL. A: Original inputs; B: Level 0 (L0)—absent RL (only UL); C: Level 1 (L1)—low RL; D: Level 2 (L2)—moderate RL; E: Level 3 (L3)—high RL; F: Level 4 (L4)—extreme RL (only RL).

mechanisms of RL [18], based on (1) an initial generation of noisy and stochastic representations, (2) a slow improvement in the prediction of the future reward (surprise) and (3) a representation learning based on both the stochastic generation and the surprise. Instead, the initial phases of the UL training can proceed regardless of the slow learning to predict future reward (success of behaviour), and at the same time building suitable representations for the behaviour itself. However, with the advancement of training the conditions with absent RL (L0) and low RL (L1) achieve a lower performance than the more balanced conditions (S3–S5 Figs in S1 text in S1 File—confirm the generality of this result). This phenomenon occurs because in the middle and last phases of training the other models (L2, L3 and L4) overcome the initial unstable phase, exploiting a higher task-directed bias (reward level) on the internal representations. Instead, the models with a low RL continue to encode both the task-independent and task-directed features without any specific bias. This unsupervised representation learning process is 'agnostic' with respect to the task performance and therefore causes a resources competition preventing the full exploitation of resources for task-directed computations.

At the opposite side of the spectrum, also models with an exclusive RL (extreme RL; L4) have computational limitations, resulting in sub-optimal performance (Fig 6 and Table 1). As also discussed above (Fig 5), the reconfiguration of the synaptic strengths is influenced by stochastic noisy activations and a slow reward prediction improvement. A consequence of these features is that these models show an inefficient initial representation learning, potentially incurring in local minima (e.g. Fig 9E).

These results are reproduced also by tests where we manipulated the computational resources that were available for the perceptual component (Table 2). These tests demonstrate that also with a higher amount of computational resources the best performance is achieved by the models having a balanced integration of UL and RL (moderate RL; L2). Interestingly, in the case of higher resources, the L2 model (moderate reward) shows the best performance while in the case of low resources the L3 model (high reward) shows the best one. Although this difference is small, there could be a functional explanation. Higher resources allow the model to encode more information helping to execute a correct categorisation. In particular, increasing the computational resource of the UL mechanisms lead to storing both more task-directed and task-irrelevant features, thus needing a minor reward-based bias to tune the scarce resource toward task-directed feature (low resource condition). Nevertheless, 'storing all the information without a bias toward the useful one' remains an inefficient computational strategy due to a residual competition between task-relevant and task-irrelevant features. Hence, the L0 and L1 models show sub-optimal performance also in the case of high resources. These results suggest that also in the case of high resources a trade-off between computational resource and task-directed bias leads to the best performance.

Our results fit with the experimental evidence regarding the role of feedback signals in human adaptive behaviours. For example, [32] suggest that in autistic people there could be abnormal sensory processing. Corroborating experimental evidence [91], our results support the idea that in autism the feedback processing may be diminished, causing a certain level of autonomy of perceptual learning processes with respect to the task-dependent feedback. On the other hand, abnormal reward sensitivity is considered one of the core factors bringing to clinical conditions such as drug addiction or autism [92, 93]. Indeed, [15] proposes that autistic peoples could show an imbalance toward the reward-based plasticity, causing deficits in categorisation performances (e.g. low generalisation skills). Our results corroborate this proposal, namely autistic people could show an excessive feedback-dependent sensory processing that causes sub-optimal performance with a potential loss of generalisation skills. Interestingly, the proposals of [32] (feedback insensitivity) and [15] (excessive feedback-dependent sensory processing) seem in opposition. Future investigations could clarify this controversial evidence,

however, our results agree that (1) both sides of the imbalance can be detrimental to the categorisation task performance, and (2) the two imbalances could identify different categorisation profiles of the autism spectrum conditions.

**4.1.2 Unsupervised learning, reinforcement learning, and categorical perception.** The main result presented here is that different UL/RL interactions have a different impact on the clustering process of internal perceptual representations, in turn leading to specific advantageous or disadvantageous effects on the task performance. In particular, in case of a balanced mix of the two learning processes (graphs 'C, D' of Figs 7–9) a beneficial categorical perception effect emerges. Indeed, in this case the distances between input representations associated with a specific response are reduced, while the distances between those associated with different responses are expanded. This effect is made evident by the graphical reconstruction of original inputs (Fig 11). In case of a balanced mixed of learning processes (e.g. graphs 'D' of the figure) the sensory system perceives the input as prototypes depending on the salient category (e.g., a coloured blob, when the task requires a colour-based categorisation, or a colourless shape prototype when the task requires a shape-based categorisation).

These results corroborate the functional hypothesis proposed in the previous section. In particular, a balanced mix of unsupervised and reinforcement learning lead the internal representations to be clustered according to the task demands (categorical perception), thus improving action selection without losing salient information. Furthermore, the results are coherent with scientific evidence regarding the modulations of perceptual representations. For example, [94] detect a training-dependent alteration of objects representation in human extrastriate cortices and [95] detect a motor-related modulation of extrastriate cortices (in particular the extrastriate body area). In addition, [96] report that the solution of a category learning task causes the emergence of category-based representations. This phenomenon has been also shown in mice [97] and primates [98–100], thus indicating to have a key role along with the evolution of mammal perceptual systems.

Our model supports the investigation of imbalanced perceptual learning processes, leading to an absent or dysfunctional categorical perception. For example, in the case of absent or low RL (graphs 'A-B' of Figs 7–9) the unsupervised learning mechanisms lead to the acquisition of an high amount of visual features independently of their relevance for the task. This result is confirmed by the low reconstruction error obtained by these models (Fig 10), suggesting that they store a higher amount of visual information. Moreover, the input reconstructions are very similar to the original inputs (graphs 'A, B' of Fig 11) confirming a very low loss of information. Interestingly, these models show a certain level of clustering effect on the basis of the colour category due to the visual input coding. In particular, the UL mechanisms tend to extract the most prominent statistical regularities and the colour coding is the most distinguishable feature of the inputs (as many pixels code the colour of inputs while a few pixels differentiate the same-coloured shapes of blue circles and squares). Overall, these results agree with the functional hypothesis proposed in the previous section, for which task-independent perceptual representations can cause sub-optimal performance. Moreover, the emergence of a task-independent clustering effect could worsen the perceptual representation learning process.

At the opposite side of the spectrum, in the models with an extreme reward-dependent learning (graph 'E' of the Figs 7–9) the internal representations collapse to four specific ones, depending on the task demands. The highest reconstruction error (Fig 10) confirms an extreme information loss, sometimes causing clustering errors (see graph 'E' of Fig 9). Moreover, the input reconstructions (graphs 'F' of Fig 11) offer further evidence of the strong information loss. Indeed, the model can produce task-directed representations but they look less distinguishable with respect to those of the graphs 'D and E', sometimes collapsing in a unique

task-independent representation. Although in this case the reward signal can support a task-dependent clustering effect, these models show a sub-optimal performance. This corroborates the idea that extreme reward-based learning can give a general advantage to a perceptual system but it can also cause clustering errors. As detailed in the previous section, these disadvantages are caused by a slower and more variable learning mechanism of RL. Moreover, in this case the UL/RL imbalance can cause a loss of useful information, potentially getting worse generalisation skills.

These results could explain the proposals of [32] and [15], suggesting that a weak top-down signal or extreme RL plasticity could affect categorisation and generalisation skill in autistic persons. Overall, the results we extracted from the 'extreme cases' could explain the altered computation in sensory cortices of autistic persons [29, 31]. Indeed a recent review [30] proposes that an altered sensory computation in the visual cortex is a key aspect to building better models of autism spectrum disorders. As explained in the next section, future investigations could clarify these experimental evidence.

## 4.2 Main contributions, clinical relevance and technological implications

Overall, our results propose many insights into the learning processes leading to categorical perception. First, a balanced contribution of unsupervised and reinforcement learning in high-order stages of a perceptual system leads to the best categorisation performance. This advantage is supported by a categorical perception effect, for which the perceptual system stores the visual information both on the basis of statistical regularities of inputs and task-dependent salience features. Second, the extreme cases of unsupervised and reinforcement representation learning lead to suboptimal performances. In particular, exclusive unsupervised learning is inefficient due to an excessive autonomy of sensory computations with respect to the task demands. Instead, exclusive reinforcement learning causes a slow and variable sensory computation potentially leading to local minima of performance or clustering errors. These sub-optimal performances are caused by different alterations of perceptual representation learning. Indeed, in the first case the perceptual component stores too much information and hence shows a low task-directed CP effect. Conversely, in the second case the perceptual component acquires less distinguishable representations showing a maladaptive extreme information loss.

The integration of our computational approach with specific experimental protocols, focusing on the feedback effect [39], and neuroimaging techniques, supporting the investigation of task-dependent sensory representations [94, 95], could clarify the role of reward signals in healthy and clinical conditions of categorical perception. In particular, our model provides functional hypothesis and predictions about behavioural and imaging evidence. Indeed, our results suggest that the altered categorisation performance in autism could be explained by an unstable categorical perception effect in extrastriate cortices, leading to sub-optimal generalisation skills and altered sensory computations [29–31]. For example, the 'extreme unsupervised learning' model, showing a maladaptive excessive autonomy between task demands and perceptual representation learning processes, corroborates a theoretical proposal explaining the altered categorisation process in autism [32, 91]. However, the 'extreme reinforcement learning' model, reaching sub-optimal performances and potentially low generalisation skills, corroborates an alternative theoretical proposal for which autism could be supported by an extreme and inefficient reward-dependent representation learning [15]. Considering that autism spectrum condition shows many phenotypes in the social domain (e.g., iper-social and ipo-social profiles; [101]), our model reconciles the two opposing views suggesting that both the extreme UL/RL models corroborate the existence of different categorisation profiles in autism spectrum condition.

The computational principles and algorithms we used here can give a prompt to the machine learning and robotics fields. The field of reinforcement learning has a long tradition of studies that approaches the representation learning issue [83, 102], also integrating UL and RL approaches [103–105]. On the other hand, machine learning works propose many alternative architectures that aim to solve the representation learning issue. For example many studies adopt a variational auto-encoder [106] also with practical applications (VAE; [107, 108]). Moreover, recent approaches propose new variants of VAE such as the C-VAE [109], approaching a multimodal representation learning framework, or the TD-VAE [110], facing sequential representational learning. Here we used a Deep Belief Network, composed of two Restricted Boltzmann Machines, that executes a representation learning and a dimensional reduction. We adopted this network due to specific computational and bio-inspired features. First, VAEs are commonly implemented with Gaussian units while CD and REINFORCE are both natively implemented with Bernoulli units. This feature has allowed us to easily integrate the two algorithms in a single training equation of the DBN. Second, during training the error back-propagation influences each layer of the VAE while the DBN can be trained in a layer-wise way and each layer can be trained with a different algorithm. These features have allowed us to adopt different CD-REINFORCE balances along the DBN hierarchy, hence emulating the different impact of the reward at different stages of the brain sensory system. Third, CD and REINFORCE show a localistic learning rule that allows us to keep a certain level of bio-plausibility with respect to the error back-propagation algorithm [27]. Despite these features, ML approaches start to integrate many learning mechanisms to improve the efficiency of the representation learning process [33, 34]. For example [111], propose a first approach to train a VAE with a training function that integrates the error back-propagation algorithm with a secondary object function that potentially supports a reward signal. Future studies could explore the possibility to compare our DBN, trained with our novel algorithm, with a VAE trained with both error back-propagation and RL.

In addition to the previous studies, recent advances in deep learning [35, 36] and deep reinforcement learning [112] are starting to elaborate indices to evaluate the task-related efficiency of representations, also investigating the issue of categorical perception in deep neural networks [35]. Taking inspiration from the different brain processes that support representation learning in healthy and clinical human conditions, our approach can serve as a guide for these ML studies. For example, by analysing the categorisation deficits affecting humans in clinical conditions (e.g., autism) we could identify the latent causes that lead to generalisation limits in deep learning. On the side of robotics, some approaches [113–115] start to create learning functions integrating unsupervised learning and task-dependent reward functions, with the aim of better discriminating the visual features that provide the robot with better control on the environment. Overall, our approach could prompt the construction of new robotic architectures, taking advantage of a balance between agnostic and task-directed perceptual processes [116, 117].

## 4.3 Other computational models of categorical perception

The computational literature concerning perceptual and learning processes is vast, involving many fields such as perceptual decision making, perceptual learning, category learning. Here we focus on categorical perception and we compare our model with other recent models that explicitly investigate this phenomenon (see [118] for a previous review of the categorical perception models).

The work [25] proposes an evolutionary approach to model categorical perception effects. In this work, an embodied agent, supported by a recurrent neural network and genetic

algorithms, shows embodied loops with the world and evolves internal representations that support categorisation processes (embodied categorical perception). Despite the strong methodological differences with our proposal (e.g., the use of genetic algorithms), we share the interest in system-environment interactions and perceptual realism of the input leading to the emergence of categorical perception.

The work [19] proposes a computational model of perceptual learning processes and categorical perception. The authors build a bio-grounded architecture showing a functional differentiation between computations in apical dendrites (top-down feedback-dependent inputs from other regions, e.g. linguistic or attention processes) and basal dendrites (bottom-up sensory-driven inputs from sensors). Emulating the inter-cortical interaction, the unsupervised learning occurs at different stages of visual hierarchy and leads to the emergence of a categorical perception effect. The model shares with our proposal the idea that categorical perception is supported by an integration between bottom-up signals (input-driven) and top-down signals (feedback-driven). However, this model supports this integration trough a bio-plausible hardwired connectivity while our proposal exploits a novel learning rule emulating the integration of associative and reward-based signals in the brain [16].

The work [20] proposes a model of speech production showing a categorical effect. The model shows a neuro-inspired system-level architecture that includes many cortical and subcortical modules (e.g., sensory, motor and linguistic layers) and it is trained trough an unsupervised learning rule (self-organising maps; SOMs). Similar to ours, this model adopts a system-level modelling approach that aims to emulate many cortical and subcortical functions. However, the proposal adopts a pure unsupervised learning rule to train the weights between the layers while our proposal involves both unsupervised and reinforcement learning mechanisms. This allows us to better investigate how task demands affect the organisation of internal representations.

The work [21] proposes a computational model that emulates the acquisition of categorical perception in infant human auditory systems. In particular, they produce many ecological inputs (vowel sounds) and adopt a bio-plausible Hebbian SOM (unsupervised learning; UL). As in our work, the authors used realistic inputs to emulate the sensory processes. This solution improves the interpretability of the internal representations on the basis of more 'ecological features' (e.g. vowel sounds or RGB pixels). However, the authors manipulate the input pattern frequencies to bias the SOMs for inducing the representation of prototypical categories. Instead, we used a set of input patterns with the same frequency and the model nevertheless acquires the representation of prototypical categories. Moreover, the authors adopt a pure UL rule while our model is trained with a novel rule that integrates both UL and RL.

The work [22] proposes a bio-plausible model reproducing the emergence of categorical perception in the brain visual system. The system is composed of three sequential layers, of which the first encodes low-level visual features and the last receives both from the previous ones and from an external top-down source. This last top-down input causes the category learning. Each layer implements a competitive mechanism based on lateral inhibition and the whole architecture learns through a bio-plausible unsupervised Hebbian learning rule. Similarly to our work, this model proposes a hierarchical visual system (composed of different sequential computation levels) and adopts ecological inputs to train the model. However, the model emulates the visual hierarchy abstracting other brain structures, namely the top-down feedback input is completely abstract while in our model it depends on many modules of a motivational system. Moreover, the model exploits an unsupervised learning rule while ours considers also reinforcement learning to encode the feedback. Last, the feedback mechanisms of the model only influence the top-layer while in our case the RL-based feedback biases both the top motor layer and the intermediate perceptual level.

**Table 3. Overview of the main features of the computational models on categorical perception considered here.**

| Models | Computational features | | | Bio-plausible features | |
|---|---|---|---|---|---|
| | Algorithms | Learning mechanism | System-level approach | Architecture | Learning processes |
| Beer (2003) [25] | Recurrent network | (Genetic algorithm) | ✗ | ✗ | ✗ |
| Spratling and Johnson (2006) [19] | Bio-constrained network | Unsupervised | ✔/✗ | ✔ | ✔ |
| Kröger et al. (2007) [20] | SOMs | Unsupervised | ✔ | ✔ | ✔ |
| Salminen et al. (2009) [21] | SOMs | Unsupervised | ✗ | ✗ | ✔ |
| Casey and Sowden (2012) [22] | Bio-constrained network | Unsupervised | ✔/✗ | ✔ | ✔ |
| Tajima et al. (2016) [24] | RNN | (Bayesian inference) | ✗ | ✗ | ✔/✗ |
| Pérez-Gay et al. (2017) [23] | Autoencoder + MLP | Unsupervised, Supervised | ✗ | ✗ | ✗ |
| **Granato et al. (2021)** | Actor-Critic, Deep Belief Network, auxiliary components | Unsupervised, Unsupervised/ Reinforcement | ✔ | ✔ | ✔ |

SOMs stands for self-organising maps; 'MLP: Multi-layer perceptron. Entries in brackets under the respective column are not proper 'Learning mechanisms'. 'System-level approach' indicates whether the model emulates the computations of many brain structures beyond the perceptual component (e.g. subcortical structures). 'Bio-plausible features' indicates whether the model captures some aspects of the brain architecture (e.g., functioning of neurons and/or interactions of macro-systems) or learning processes (i.e., bio-plausible learning rules).

The work [24] proposes a computational model that emulates the neural populations dynamics during the acquisition of colour-based categorical perception. The model is supported by a simple recurrent neural network composed of a sensory and a category layer, in which a Bayesian inferential top-down process allows the second layer to influence the lower one with respect to categorical encoding. Despite this proposal adopts a neuro-inspired approach, it has marked differences with respect to our work. The model does not use a true 'learning process', in that the emergence of categorical perception is based on a top-down inferential process. In this sense, the model has common features with our previous works on representation manipulation [3, 37], in which a recurrent neural network biases a sensory system and leads to the emergence of categorical perception. Moreover, the inferential process is not influenced by a performance-related feedback signal. Indeed, the model does not emulate the contribution of reward signals produced by subcortical structures as our model did.

The work [23] proposes a computational model of categorical perception with which the authors investigate different learning processes. They adopt a functional approach based on machine learning, including an auto-encoder (AE; [26]) and a classifier. The model undergoes an UL phase (only VAE training) and a supervised learning phase in which the whole model (both the trained VAE and the classifier) has to categorise the input on the basis of external labels. By comparing the internal representations of the AE after the UL phase with those after the SL phase, the authors detect a categorical perception effect. Similarly to our model, the authors adopt a functional approach based on a generative model and a classifier model. Moreover, they investigate how the interaction between unsupervised and feedback-dependent phases can support the emergence of categorical perception. However, we adopt a neuro-inspired approach to build the model, showing a higher biological plausibility. Furthermore, our learning protocol involves a pure unsupervised learning phase only before the task starts, while the task performance integrates both unsupervised and feedback-dependent signals.

Table 3 shows a list of the models we have taken in consideration here. The table highlights that most models encompass learning processes, with the exclusion of [25] and [24] involving

evolutionary and inferential processes respectively. Moreover, several models adopt unsupervised learning rules. Despite unsupervised associative mechanisms have a key role in categorical perception, empirical evidence strongly points to the fact that several brain areas integrate multiple learning processes (i.e., supervised, unsupervised, reinforcement, [16]). Interestingly no model on categorical perception integrates reinforcement learning mechanisms, while our proposal shows both an unsupervised phase and an integrated unsupervised/reinforcement phase. At last, with the exclusion of [20] and our proposal, the bio-plausible models tend to focus on a particular brain system while abstracting the computations of other structures (system-level approach).

## 4.4 Limits and future directions

Although the previous section shows the advancements of our model with respect to the others, it has still some limitations we intend to overcome in future work, together with the development of other interesting aspects of the model. We discuss the main ones in this section.

**4.4.1 Bio-plausibility and neuro-inspired/bio-grounded approaches.** The computational model presented here is supported by a neuro-inspired architecture in which the key components are implemented with neural networks (a generative neural network and an actor-critic network). Although the architecture is in functionally inspired by the interaction between cortical and subcortical brain systems (sensory-motor cortices and basal ganglia) and has a certain level of bio-plausibility (e.g., localistic learning rules; [27]), the model is based on simplified neurons and abstract plasticity rules. Future work could aim to develop the ideas proposed here on the overall architecture and components' interactions by using neural networks having a higher degree of biological detail. For example, we could build models based on spiking neurons and bio-grounded learning rules such as STDP [119, 120] but integrating plasticity rules that involve a reward signal as done in [121]. Moreover, we could use spiking generative models [122–124] to emulate the STDP effects on representation learning processes. These implementations would support further investigations about brain plasticity and the emergence of categorical perception.

**4.4.2 Data fitting and model updates.** We qualitatively compared the perceptual processes of the architecture proposed here with the experimental evidence in healthy and clinical conditions. However, the comparison was only a proof-of-concept and the model needs to be tested against detailed experimental data. To overcome this limitation, we aim to enhance the motor component of the model, for now representing a simplified output, making it able to produce performances comparable to those of humans as we did in [3].

**4.4.3 Embodiment and robotic environment.** We will also aim to follow a second complementary direction of research moving towards neuro-cognitive robotics by linking the whole architecture to a robotic arm. This approach would allow the reproduction of human motor movements during a sorting task, supporting the investigation of cognitive processes underlying category learning. A robotic arm would allow the architecture to autonomously develop more complex embodied processes. For now, the model emulates only some essential elements of embodiment (e.g. realistic sensory input; an environment feedback, based on the model performance, that influences the model perception) but a simulated or physical robotic environment would support investigations on the relationships between categorical perception, motor skills, and embodiment [46–48], also in the case of clinical conditions (e.g., autism; [125]).

**4.4.4 Transfer learning skills and generalisation analysis.** Here we consider three category learning tasks in isolation and a different model solves each one of the three task conditions (either sorting rule for colour, or shape, or size). We adopted this strategy due to the large amount of computational resources required to systematically study the multiple learning

conditions (i.e., three sorting rules, five RL levels, two resource levels of perceptual component). In particular, we repeated the task for each of the thirty conditions for a total of over 1000 simulations. This approach allowed the execution of robust statistical analyses but it prevents testing the usefulness of representations acquired in a single task condition for the solution of the other two task conditions (i.e., transfer learning; [126–128]). Moreover, our approach prevents the investigation of adaptive categorical perception, for which the model is required to further adapt its perception in case the sorting rule unexpectedly changes during the task performance. To overcome these issues, we aim to test the model with two further task conditions. First, we could implement a 'static generalisation condition' in which the model is tested with other categories after the 'principal task', keeping fixed the perceptual component. This test should clarify the relationship between the UL/RL balance and the generalisation skills of the perceptual component. Second, we could implement an 'adaptive categorical perception condition' in which the sorting rule suddenly changes many times and the model has to adapt online its perception and responses to the new requests. This test should clarify the relationship between the UL/RL balance and the perceptual adaptation of the model. Overall, we expect that the extreme RL model (L4), producing the most task-directed representations, might lose the generalisation and adaptation capacities due to its extreme information loss impacting on the task-independent features. Conversely, the models with a more balanced RL/UL ratio (L2 and L3), could show the best performances both in the main sorting task (as shown here) and in these two new tasks. This would corroborate the idea that a balanced UL/RL mix is the most suitable solution for an artificial and biological perceptual component, needing to adapt to an uncertain environment where the task can change (e.g. novel objects to categorise) and the computational resources are limited.

**4.4.5 Multi rules categorisation and catastrophic forgetting.** The model is able to adapt its motor, motivational, and perceptual components to solve a sorting task that shows a fixed single sorting rule (sorting for colour, shape, or size). Although the system could slowly adapt after a rule change, it would likely incur into catastrophic forgetting (i.e. the loss of the already acquired information caused by the acquisition of new ones; [129, 130]). This limitation, strongly linked to the previous ones, is due to the fact that an ideal perceptual system should be able (1) to transfer the knowledge to another task or task condition (transfer learning) without losing the previously acquired information (catastrophic forgetting) and (2) to quickly adapt itself in case the initial sorting rule changes (adaptive categorical perception). To overcome this limitation we could integrate the architecture presented here with mechanisms implementing an internal manipulation of perceptual representations as studied in our recent computational models [3, 37]. In those models, a dynamical working memory encodes different categorisation rules and guides an internal 'top-down manipulator' that selects different portions of a visual neural network. The integration of this internal manipulation and the learning processes studied here should allow an architecture to select and train specific portions of a neural network, improving the problem of catastrophic interference and quick perceptual adaptation (e.g., 'experts approach'; [128]).

**4.4.6 Category learning, categorical perception, and perceptual learning: Differences and model updates.** The proposed architecture focuses on CP in case the task-directed actions (e.g. category learning) alter the perceptual representations (differences and similarities expansion). On the other hand, 'perceptual learning' refers to the 'experience-dependent enhancement of our ability to make sense of what we see, hear, feel, taste or smell' [131]. Interestingly, the work [2] suggests that category learning and perceptual learning could share specific learning mechanisms, as in the case of the emergence of categorical perception. Despite these commonalities, controversial evidence highlights some differences between these processes. For example, it is not clear if category learning and perceptual learning influence the

perceptual systems at the same level (early, middle or late processing stages). To clarify the controversial evidence, we could extend our investigations by executing specific model updates. For example, we could apply the same learning rule, integrating UL and RL, in each sensory-motor hierarchy of our networks. In particular, in addition to the second RBM of the DBN (from the first hidden layer to the second hidden layer of DBN), we could apply the same RL/UL rule on the first RBM (from the input layer to the first hidden layer of DBN). In this way we could potentially set different levels of UL/RL integration at each level of abstraction (from the low-level perceptual processes to the motor selection). Searching for the model configurations that best fit the human data, we could investigate the differences in learning processes supporting category learning and perceptual learning, in particular the reward/task influence at different levels of abstraction.

## 5 Conclusions

In this work we investigated how the interaction between unsupervised and reinforcement learning leads to the emergence of human categorical perception. We integrated neuroscientific evidence and machine learning methods (e.g. generative neural networks) to build a neuro-inspired computational model that is able to perform a category learning task. In particular, the system-level architecture shows neuro-inspired components (emulating cortical and sub-cortical brain functional macro-systems) and integrates bio-plausible unsupervised and reinforcement learning processes (e.g., distributed representations and localistic learning rules). The analyses of internal representations and performance suggest that a balanced mix of unsupervised and reinforcement learning supports the acquisition of suitable task-directed representations (categorical perception), leading to the best performances. Instead, extreme cases lead to sub-optional performances due to maladaptive representation learning processes. In particular, in the case of limited computational resources the models without reinforcement learning are not able to focus on relevant features thus producing sub-optimal performances. Instead, the models without unsupervised learning show more unstable and slow learning processes, especially at early phases of learning, thus incurring in clustering errors and an excessive loss of information. The model qualitatively reproduces experimental evidence in healthy conditions, namely the emergence of category-based representations in extrastriate cortices. Moreover, the model can explain the altered categorisation performance in clinical conditions such as autism. For example, the model with only unsupervised learning shows an excessive sensory autonomy with respect to the task-dependent feedback, possibly explaining the worse categorisation processes in some autism conditions. Moreover, the model with only reinforcement learning explains the low generalisation skills in some other autism conditions, due to an excessive loss of information. These opposite effects can explain the heterogeneity of autism spectrum conditions, as a different imbalanced mix of unsupervised and supervised learning mechanisms in different autistic people. The model could also support the development of machine learning systems able to undergo categorical perception effects, and robotic systems needing to face uncertain environments through suitable representations. In particular, our neuro-inspired algorithm could prompt the development of new algorithms that are able to autonomously balance UL and RL processes depending on the task demands, the available computational resources, and generalisation requirements.

## Supporting information

**S1 File. S1 text (Supplementary materials).** We insert in this file many details regarding the model implementation (networks layers, hyper-parameters, task conditions, etc) and the

computation of the results.
(PDF)

## Acknowledgments

We thank the Neuroscience Gateway [90] used to run most of the simulations.

## Author Contributions

**Conceptualization:** Giovanni Granato, Emilio Cartoni, Federico Da Rold, Andrea Mattera, Gianluca Baldassarre.

**Data curation:** Giovanni Granato, Emilio Cartoni, Federico Da Rold, Andrea Mattera, Gianluca Baldassarre.

**Formal analysis:** Giovanni Granato, Emilio Cartoni, Federico Da Rold, Andrea Mattera, Gianluca Baldassarre.

**Funding acquisition:** Gianluca Baldassarre.

**Investigation:** Giovanni Granato, Emilio Cartoni, Federico Da Rold, Andrea Mattera, Gianluca Baldassarre.

**Methodology:** Giovanni Granato.

**Project administration:** Gianluca Baldassarre.

**Software:** Giovanni Granato.

**Supervision:** Emilio Cartoni, Gianluca Baldassarre.

**Validation:** Giovanni Granato, Federico Da Rold, Andrea Mattera.

**Visualization:** Giovanni Granato.

**Writing – original draft:** Giovanni Granato.

**Writing – review & editing:** Giovanni Granato, Emilio Cartoni, Federico Da Rold, Andrea Mattera, Gianluca Baldassarre.

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
