## [Decision Letter · Decision Letter 0]

2 Feb 2022

PONE-D-21-40171Integrating unsupervised and reinforcement learning in human categorical perception: a computational modelPLOS ONE

Dear Dr. Granato,

Thank you for submitting your manuscript to PLOS ONE. After careful consideration, we feel that it has merit but does not fully meet PLOS ONE’s publication criteria as it currently stands. Therefore, we invite you to submit a revised version of the manuscript that addresses the points raised during the review process.

We look forward to receiving your revised manuscript.

Kind regards,

Frederic Alexandre

Academic Editor

PLOS ONE

Journal Requirements:

2. We note that you have included the phrase “data not yet published” in your manuscript. Unfortunately, this does not meet our data sharing requirements. PLOS does not permit references to inaccessible data. We require that authors provide all relevant data within the paper, Supporting Information files, or in an acceptable, public repository. Please add a citation to support this phrase or upload the data that corresponds with these findings to a stable repository (such as Figshare or Dryad) and provide and URLs, DOIs, or accession numbers that may be used to access these data. Or, if the data are not a core part of the research being presented in your study, we ask that you remove the phrase that refers to these data.

Additional Editor Comments:

Both reviewers have commented on the paper and advise a major revision. More specifically, claims in the abstract and introduction should better correspond to what is reported in the Limitations part. In addition and as requested by the reviewers, some details about the model should be added for clarity. Thank you for considering these points.

Reviewers' comments:

Reviewer's Responses to Questions

**Comments to the Author**

1. Is the manuscript technically sound, and do the data support the conclusions?

Reviewer #1: Partly

Reviewer #2: Partly

2. Has the statistical analysis been performed appropriately and rigorously? 

Reviewer #1: Yes

Reviewer #2: Yes

3. Have the authors made all data underlying the findings in their manuscript fully available?

Reviewer #1: Yes

Reviewer #2: Yes

4. Is the manuscript presented in an intelligible fashion and written in standard English?

Reviewer #1: No

Reviewer #2: Yes

5. Review Comments to the Author

Reviewer #1: In the present article, the authors propose a model for human categorical perception and the underlying learning process. They train a multi-layer artificial neural network to perform an image-sorting task. In this framework, they show that a combination of Hebbian unsupervised learning and reward-driven learning in a given layer of the network is significantly more efficient (better final performance) than Hebbian learning alone or reward-driven learning alone.

While the main result of the paper may be interesting, the incomplete description of the model’s implementation makes it difficult to understand the breadth of the results or to reproduce it. The conditions in which the model is tested are not very well described and it is difficult to know if they can be generalized to other tasks and/or representations of the categorical learning system representation. Finally, the model of the brain networks involved in categorical perception is quite sophisticated (many layers and several learning algorythms in different layers) while the representation of elementary computational units is simplistic and based on previous frameworks developed in machine learning. It is thus unclear how the articulation between various components of the artificial network may be linked with natural brain processes underlying categorical perception, as suggested by the title and abstract.

Overall, I would be more confident in my opinion on this work if the model’s implementation, the task and the results were described in full details. Moreover, the results of the model could be discussed in relation to brain processes in the discussion rather than being interpreted in terms of categorical learning in brain networks in the abstract and title.

Specific comments:

Abstract:

It’s not clear if all assertions from the abstract are coming from the authors results or if they come from previous studies. In particular, the sentence “Indeed, an excessive unsupervised learning contribution tends to underrepresent task-relevant features while an excessive reinforcement learning contribution tends to initially learn slowly and then to incur in local minima.” There is no result about local minima and this is only evoked in the discussion as a possible interpretation of the results. If so, it should not stand as an assertion in the abstract.

Figure 1 : Instead of simply showing examples of visual elements, a graphical description of the task would help the reader understand what is modelled here.

Methods:

1.1: The task description is not clear. While the authors explain how the task is linked to other tasks, it is not clear what exact task is modelled here and how it is represented/run in the model.

1.2: This part does not include a description of the model is not full and it is unclear why it belongs to the methods as is not linked to the model’s implementation.

The extensive comparison between the actual model and its interpretation rather belongs to the discussion of the paper. As an example, the authors explain first how the model may be “coherent with the theoretical framework of embodied perception”, even before explaining what the model consists in. Figure 2, summarizing the parallel between the model and real-world brain structures (again rather a discussion point) stands before any description of the model. Same in the text (e.g. “The architecture and learning processes of the model are inspired by the interactions between brain cortical and sub-cortical macro-systems”). As a result, the reader is left with speculations rather than being able to understand what the authors did.

Discussing future work also belongs to the discussion rather than methods (“Future works will make the learning mechanisms of the model more gradual along the sensory-motor hierarchy”).

In this part, the authors simply refer to Figure 3 (schematics of the model’s component) and provide a qualitative description of each component, again making the hypothetical link to biology and other work rather than describing the implementation they performed (e.g. “This component is based on a neural network that receives visual inputs and performs information abstraction, mimicking the brain visual system.”).

It is not clear why Figure 3 is different from Fig 4 and why part 1.2 precedes the real methods part (1.3 describing model implementation). Why should the similarity to other works and biological network not rather be in the discussion? Because it stands before the implementation of the model, it adds rather confuses the reader in my opinion.

1.3 The model implementation is given in this part.

Still, some details are missing to fully understand the mode without going to the cited publication:

- How is the RBM ‘trained’? What is the ‘reconstructed visible activation’? Does an RBM have 2 layers or more? What is data and model (reconstructed) then? In other words, if one RBM is 2 layers (v and h), how can it have a data layer and a model layer? I did not understand. A figure representation of the component could help understanding its structure and the various names used.

- Same problem with the second RBM.

- Motivational component. The predictor module is not fully described here.

Moreover, the description is hard to follow due to the lack of figure.

- Motor component: is it an RBM as well? Or a single layer? A representation would help

- A global view of the network with all layers and names would help the reader follow the model’s description and implementation.

Results:

Just as the methods section, the results section lacks precision. As an example, it starts with “We tested the model with different conditions of the sorting task, each involving one out of three sorting rules. For example, …”. The different conditions are never described unambiguously (unless I missed it).

Another example in the following sentences: “Note that 50 units were sufficient to allow the system to fully encode the image features, as shown by a preliminary test indicating a close-to-null reconstruction error.” This result is not shown anywhere (neither the result of the reconstruction, nor the error value, nor how the error is computed).

The clear and full description of the task begin still missing, it is difficult to understand the beginning of the results description.

The results talk about performance in groups not being defined before.

Discussion:

The 10 pages discussion appears a bit excessive given the relatively thin results section and the application of the learning algorithm in a very tight context.

Reviewer #2: The authors present a systems-level categorical perception model and present computational simulation results establishing that a balanced mix of unsupervised and reinforcement learning leads to emergence of categorical perception. The authors present a simple system level approach that introduces a balancing of unsupervised learning with reinforcement learning, and as the authors claim, this could be a proof-of-concept architecture that could give clinical insights in autism. The paper is also well written and explains the architecture of the model and its limitations well.

The following are the major concerns:

1. The authors put forward the argument that dopamine/RL is involved in categorical perception? Is there any empirical evidence to back up this claim?

2. Though the task is adaptive categorical perception, the paper reduces it to a clustering study and shows representations that are optimal for each task in isolation. There is no adaptation in the algorithm itself and it does not solve these tasks simultaneously. The time scales of the reinforcement learning where it is learnt though a policy gradient for each task does not permit rapid adaption and the authors point out this in the limitations section that it would cause catastrophic forgetting. The reader does not know until the limitations section that the proposed model/simulations do not address the adaptive categorical perception. Hence the abstract, motivation, introduction could be misleading to the reader who might be eagerly waiting for n algorithmic solution to the adaptive categorical perception. This needs to be spelt out before the limitations. This can be done in two ways: i) the computational model details as well as preliminary solution for the adaptive version could be included or ii) the manuscript could focus on the categorical perception solution with balanced UL+RL and present the adaptive version as a possible extension as a separate section in the discussion. The authors could decide. Not presenting the model/simulations for adaptive version does not mitigate the importance of the current model and results.

3. In section 1.2, it is mentioned that the architecture is coherent with the theoretical framework of embodied perception. However, I feel the motor component is not central to the model and acts as an addendum, it is not clear how this model develops a coherent link with embodied cognition other than its simplistic motor component.

4. The paper puts forward RBM as the unsupervised component, the authors could have a discussion on other possible architectures that could do adaptive categorical perception, for example a VAE or a conditional VAE. They could discuss the relative merits and demerits.

Re: Karol Gregor, George Papamakarios, Frederic Besse, Lars Buesing, Theophane Weber:

Temporal Difference Variational Auto-Encoder. ICLR 2019

5. The paper claims that this intermixing of UL & RL in the work could lead to more adaptive architecture and is relevant to machine learning. Recent works on this could be explained in a section but it also has to be pointed out RL has a long history with representation learning and interaction of UL& RL.

6. The paper puts forward the argument that the interaction of UL & RL is necessary for adaptive categorical perception and other sensory processing and rules out the possibility that adaptive representations will not arise through RL alone. Of course some of these labels are semantics but there have been architectures driven purely by reward related learning that have shown complex adaptive processing. For example. the meta RL as a prefrontal cortex paper (Ref below) arrives at adaptive dynamics purely by reward related feedback. It might be misleading to point out interaction between UL+RL is the only way for adaptive perception to take place and it could do with the simplified architecture of the model.

Ref: Wang, J.X., Kurth-Nelson, Z., Kumaran, D. et al. Prefrontal cortex as a meta-reinforcement learning system. Nat Neurosci 21, 860–868 (2018). https://doi.org/10.1038/s41593-018-0147-8

Additional references that the authors could consider:

David Ha, Jürgen Schmidhuber 2018). World Models. CoRR abs/1803.10122.

Doya, K. (1999). What are the computations of the cerebellum, the basal ganglia and the cerebral cortex? Neural Networks, 12(7-8), 961–974. https://doi.org/10.1016/S0893-6080(99)00046-5

Minor editorial suggestion:

Authors refer to a 2D input stimulus as "parallelopiped", perhaps "rectangle" is more appropriate?

6. PLOS authors have the option to publish the peer review history of their article (what does this mean?). If published, this will include your full peer review and any attached files.

Reviewer #1: No

Reviewer #2: **Yes: **Raju Surampudi Bapi

---

## [Author Response · Author response to Decision Letter 0]

8 Mar 2022

We have attached the PDF file with the replies.

---

## [Decision Letter · Decision Letter 1]

18 Apr 2022

Integrating unsupervised and reinforcement learning in human categorical perception: a computational model

PONE-D-21-40171R1

Dear Dr. Granato,

We’re pleased to inform you that your manuscript has been judged scientifically suitable for publication and will be formally accepted for publication once it meets all outstanding technical requirements.

Kind regards,

Frederic Alexandre

Academic Editor

PLOS ONE

Additional Editor Comments (optional):

Thank you for taking into account the remarks given by the reviewers. This results in a much precise and clear paper !

Reviewers' comments:

Reviewer's Responses to Questions

**Comments to the Author**

1. If the authors have adequately addressed your comments raised in a previous round of review and you feel that this manuscript is now acceptable for publication, you may indicate that here to bypass the “Comments to the Author” section, enter your conflict of interest statement in the “Confidential to Editor” section, and submit your "Accept" recommendation.

Reviewer #1: All comments have been addressed

Reviewer #2: All comments have been addressed

2. Is the manuscript technically sound, and do the data support the conclusions?

Reviewer #1: Yes

Reviewer #2: Yes

3. Has the statistical analysis been performed appropriately and rigorously? 

Reviewer #1: Yes

Reviewer #2: Yes

4. Have the authors made all data underlying the findings in their manuscript fully available?

Reviewer #1: Yes

Reviewer #2: (No Response)

5. Is the manuscript presented in an intelligible fashion and written in standard English?

Reviewer #1: Yes

Reviewer #2: Yes

6. Review Comments to the Author

Reviewer #1: The authors have answered most of my comments issued in the previous round of review and I have no further comment.

Reviewer #2: The authors have now revised manuscript addressing all the concerns raised as best as they could. I am pleased with the revised version. The abstract and the main manuscript align better now. The added detailed figure of the architecture in the Supplementary makes the details clearer. The details of the parameters and the values used in various components of the architecture (provided in Supplementary) make it easier to reproduce the results. The discussion is structured better and is apt.

7. PLOS authors have the option to publish the peer review history of their article (what does this mean?). If published, this will include your full peer review and any attached files.

Reviewer #1: No

Reviewer #2: **Yes: **Raju Surampudi Bapi

---

## [Editor Report · Acceptance letter]

20 Apr 2022

PONE-D-21-40171R1 

Integrating unsupervised and reinforcement learning in human categorical perception: a computational model 

Dear Dr. Granato:

I'm pleased to inform you that your manuscript has been deemed suitable for publication in PLOS ONE. Congratulations! Your manuscript is now with our production department. 

Kind regards, 

on behalf of

Dr. Frederic Alexandre 

Academic Editor

PLOS ONE